# The Future Packaging of the Food Industry: The Development and Characterization of Innovative Biobased Materials with Essential Oils Added

**DOI:** 10.3390/gels8080505

**Published:** 2022-08-14

**Authors:** Roxana Gheorghita Puscaselu, Andrei Lobiuc, Gheorghe Gutt

**Affiliations:** 1Faculty of Medicine and Biological Sciences, Stefan Cel Mare University of Suceava, 720229 Suceava, Romania; 2Faculty of Food Engineering, Stefan Cel Mare University of Suceava, 720229 Suceava, Romania

**Keywords:** biopolymers, antimicrobial, environment, film, coating

## Abstract

The need to replace conventional, usually single-use, packaging materials, so important for the future of resources and of the environment, has propelled research towards the development of packaging-based on biopolymers, fully biodegradable and even edible. The current study furthers the research on development of such films and tests the modification of the properties of the previously developed biopolymeric material, by adding 10, respectively 20% *w*/*v* essential oils of lemon, grapefruit, orange, cinnamon, clove, mint, ginger, eucalypt, and chamomile. Films with a thickness between 53 and 102 µm were obtained, with a roughness ranging between 147 and 366 nm. Most films had a water activity index significantly below what is required for microorganism growth, as low as 0.27, while all essential oils induced microbial growth reduction or 100% inhibition. Tested for the evaluation of physical, optical, microbiological or solubility properties, all the films with the addition of essential oil in the composition showed improved properties compared to the control sample.

## 1. Introduction

The use of biopolymers in the development of packaging materials is increasingly common due to the properties they possess: biodegradability, compostability, non-toxicity, reuseability, ease of handling, the possibility of incorporating various active substances, but also relatively low costs [1,2]. Their use has been transposed to other fields, such as the biomedical one. Due to high biocompatibility, biopolymers are today successfully used in tissue engineering or wound dressing development. Numerous hydrocolloids have been used as food packaging film, with high performances in terms of the ability to withstand environments with high humidity, preserving or even improving sensory characteristics, extending shelf-life, physico-chemical, optical or mechanical properties [3,4,5]. Their edible character was another advantage in obtaining various types of films and coatings, used successfully for packaging food or coating fresh fruits and vegetables [6,7,8]. At the moment, the most used biopolymers are those based on polysaccharides (agar, sodium alginate, carrageenan, chitosan, etc.) [9], followed by those based on proteins (collagen, casein, soy or wheat protein, etc.) and lipid ones (waxes, fatty acids, acylglycerols) [10]. Since a biopolymer usually does not exhibit high physico-chemical or mechanical performances at the same time, the possibility of combining them represents the best alternative in obtaining a material with characteristics similar to plastic [11].

Research in the field has demonstrated the fact that biopolymers possess biological properties, as they are biocompatible, non-toxic, and non-immunogenic. They do not induce allergic reactions after ingestion, psychological properties, as they contribute to the stability of the products they contain and increase the shelf life of them. They also allow the transport of various solutes, gases, water vapors, and organic molecules.

Polysaccharides such as sodium alginate or agar have been intensively studied due to their synergistic character and the ability to obtain films with very good properties: resistant, flexible, with high elasticity, glossy, with a homogeneous microstructure, without pores or fissures in the composition [12].

Alginate is a marine biopolymer that can form soluble films when obtained by casting or drying, or insoluble films when crosslinked with calcium salt [13]. The biggest advantages in using alginate are the low cost of obtaining them and the multiple possibilities of use in the food and biomedical/pharmaceutical fields [14,15], due to characteristics such as good gelling and film forming properties, pH-sensitivity, mucoadhesiveness, or crosslinking capacity [16,17].

Agar is a polysaccharide extracted from red algae. In 1972, it received the status as *Generally Recognized as Safe* by GRAS and, as well as alginate, can be used in quantum statis. Almost 80% of globally produced agar is used for human consumption. Used as a film-forming component, it forms brittle material, with poor mechanical properties, moisture barrier and thermal stabilities [18]. Therefore, for use for this purpose, it is combined with other biopolymers or plasticizers [19].

Active and smart packaging are increasingly used all over the world, especially in Japan, the USA, and Australia. In Europe, their use was regulated in 2004 by Regulation EC, 1935/2004. In 2009, the EU established “Commission Regulation (EC) No 450/2009, on active and intelligent materials and articles intended to come into contact with food”. Thus, according to the European Union, active packaging systems are designed “to deliberately incorporate components that would release or absorb substances into or from the packaged food or the environment surrounding the food” [20]. If active films based on biopolymers are used in many fields, the coatings find their applicability to fruits and vegetables. Their use is important if we take into account that most losses occur during storage, handling or transport, especially due to the fact that, even after harvesting, fruits and vegetables continue their respiratory process; applying such a coating will inhibit the ripening processes [21].

Biopolymer films and coatings can be used as active delivery systems when they contain substances with an antimicrobial or antioxidant effect [22,23]. Thus, films containing gold, silver, or copper ions were effective against the development and sporulation of microorganisms, inhibiting the growth of *Staphylococcus aureus*, *Escherichia coli*, *Candida albicans*, and *Aspergillus niger* [24]. Furthermore, the physicochemical properties were improved [25]. Recently, however, the addition of essential oils in the biopolymeric matrix has attracted the attention of researchers, who intensified their studies on the development of materials with superior characteristics. Thus, essential oils embedded in films showed inhibitory activity against numerous foodborne pathogens (Table 1).

Due to the biologically active compounds, essential oils are increasingly recognized and used in the packaging materials industry. However, usually, the properties of the films change after the addition of essential oils. For example, clove oil, when embedded in films based on alginate and carrageenan, showed strong antimicrobial activity against *E. coli*, although the tensile strength of the films was lower. The inhibition zone increased directly proportional with clove oil concentration [31]. *Mostaghimi* et al. identified that the addition of clove oil to the film based on sodium alginate was more effective against *Bacillus cereus* than *E. coli* and that the properties of the material were not negatively influenced by the incorporation of essential oil [42]. Even so, if the safety of the ingested food is taken into account, it is much more important to pursue the microbial inhibition, since the properties of the films can be improved by various other additions. The antimicrobial activity of essential oils is high. They are effective against the development and proliferation of several pathogenic microorganisms (Table 2).

The current study followed the development of films based on sodium alginate, agar, and water, according to a composition that represents, at the time of writing this study, a patent proposal sent to the Romanian State Office for Inventions and Trademarks. Since specialized studies have highlighted the active nature of films with the addition of essential oil, we used lemon, grapefruit, orange, cinnamon, clove, mint, ginger, eucalypt, and chamomile oils and we tested their effect on the films’ properties. The obtained results strengthen the specialized studies and recommend their use in the development of materials with superior characteristics.

## 2. Results and Discussion

The films obtained by the casting method were observed immediately after drying and subsequently tested. All the samples that contained essential oil in the composition presented a specific taste and smell, the intensity of which varied with the volume of oil added to the film-forming solution. All the films were glossy, pleasant to the touch, soft, with regular edges and without undissolved particles. The absence of insoluble particles and fissures or pores can also be seen from the microscopic images and microtopographies obtained (Figure 1).

With the exception of the film with the addition of ginger essential oil, which broke during drying and showed extremely low flexibility, and the film with 10% mint EO, which was also less flexible, all other films were very flexible and allowed multiple bendings. The sample with 10% essential oil of mint showed high adhesiveness and can be used as a self-adhesive film. The sample with 20% essential oil of eucalyptus showed a tendency to tighten upon drying, which proves the need to reduce the addition of essential oil or increase the amount of glycerol or Tween 80. According to the images in Figure 1, the most homogeneous structure can be highlighted in the case of the film with the addition of 10% chamomile EO (**B17**). The result is also confirmed by its reduced roughness, unlike other samples-146.90 nm (Table 3).

According to the data in the table above, the mechanical properties of the films changed with the increase in the volume of essential oil. Thus, the films with 20% lemon, grapefruit, cinnamon, ginger, and eucalyptus EO presented lower values of tensile strength, unlike those with 10% addition. Increasing the volume of orange, clove, and mint essential oil favored the development of stronger films. The most resistant films were those with the addition of 10, respectively 20% mint EO (**B11**—0.274 MPa and **B12**—0.288 MPa), the one with the addition of 10% essential oil of chamomile (**B13**—0.261 MPa), and the one with 10% eucalyptus EO (**B17**—0.282 MPa). The addition of orange, cinnamon, cloves and 20% eucalyptus EO had a negative effect on the tensile strength, the values obtained being below that of the control sample (**B19**—0.135 MPa).

Similar to the resistance to breaking, the elongation of the samples with the addition of lemon oil showed reduced values. Thus, they were approximately 30% lower than the control sample (1.51, respectively 1.91% compared to 5.04%), while the film with 10% clove EO also showed lower values than the control sample. Even if some films showed lower elongation values with the increase in the volume of added essential oil, the values are still higher than the control sample (see films with lemon (**B1**, **B2**), cinnamon (**B7**, **B8**) and clove (**B9**, **B10**) EO).

The film with 10% essential chamomile EO (**B13**) presented the best mechanical properties, both tensile strength (0.261 MPa) and elasticity (26.11%). Unfortunately, the sample with 20% essential chamomile oil in the composition could not be sized according to the standards used. For the development of a film with even more improved properties, it is necessary to use a higher amount of plasticizer in the composition.

The highest roughness value (366.20 nm) was identified in **B3**—the sample with 10% grapefruit EO.

The thickness of all the films increased with the increase in the volume of added essential oil (Table 3). The biggest difference can be observed in the case of films with the addition of chamomile EO, when the thickness varied by 43.20 µm. Thus, sample **B17**, with 10% addition of EO, had a thickness of 53 µm, and **B18**, with 20% EO, had 96.20 µm. The smallest difference can be observed in the case of samples with ginger essential oil (**B13**, **B14**), when the increase in the volume of essential oil led to an increase in thickness by only 0.8 µm. The same thickness value can be observed in case of **B1** and **B2** samples, with lemon EO added. The samples with 10% orange EO (**B5**), 10 and 20% mint EO (**B11**, **B12**) and 10% chamomile EO (**B17**) had lower thicknesses than the control sample.

The values of the retraction ratio are directly correlated with those of the thickness. This means that, when it will be replicated on an industrial scale, the processor can establish the final composition of the materials knowing the retraction ratio and thickness values. The lowest retraction ratio is observed for ginger EO films (**B13**, **B14**). For all developed films, the reattraction ratio values decrease with the increase in the volume of essential oil added to the composition. This can be attributed to their hydrophobic character, so that the matrix becomes more compact and prevents the evaporation of water molecules from the composition and, implicitly, the withdrawal of the film during drying.

Except for samples **B15**–**B18**, all the films presented lower values of moisture content with the increase in the volume of EO. Compared to the control sample, most films had minor (approximately 2%) moisture content variation, except for sample B16 (20% eucalyptus), where MC was higher with 8 percentual points and samples B17 and B18 (10 and, respectively, 20% chamomile), where MC was lower with as much as 9 percentual points.

The water activity index showed similar values for the samples with the addition of lemon, grapefruit, orange, cinnamon, or clove essential oil (0.30 ± 0.03). Those with additions of mint, ginger, eucalyptus, or chamomile showed similar values (0.50 ± 0.6), but higher than the others. The control sample showed reduced values of the water activity index (0.28), similar to those of the films with the addition of clove EO (**B9**, **B10** with 0.29, respectively 0.27).

According to the images in Figure 1, the addition of essential oil had negative effects on the microstructure of the films. Thus, all samples with 10% EO volume in the composition presented more homogeneous structures than those with 20%. This may indicate the need to increase the amount of plasticizer in the composition or increase the mixing speed during the homogenization of the film-forming solution.

The transmittance of the samples varied depending on the essential oil added to the film-forming solution (Table 4). Thus, the addition of lemon, grapefruit, orange, clove, cinnamon, and ginger oil favored the develompent of more transparent films than the control sample (transmittance 68.9%). A quantity of 10% essential oil of chamomile added had the effect of obtaining a film with the highest transmittance (77%), but the addition of a further 10% essential oil favoured the development of a film with the lowest value of transmittance, 18.33%. Mint essential oil, regardless of the added concentration, had the effect of increasing the transmittance by 2.8, respectively 4.6% compared to the control sample.

The color was influenced by the type of essential oil used, although it did not vary much from the control sample. All films showed high luminosity.

The swelling index values were inversely proportional to the volume of EO added (Figure 2). This is perfectly normal if we take into account the strong hydrophobic nature of essential oils. The obtained results confirm the fact that, when it is desired to use these materials for the packaging of products with high humidity, the solubility can be improved by increasing the volume of EO added to the composition. According to Figure 2, all the samples showed a swelling tendency in the first 7 min after immersion in water (room temperature), followed by a much lower increase and even a plateau phase up to 20 min after immersion. It is very likely that the matrix will saturate and prevent additional liquid absorption. This is beneficial because the tested samples did not disintegrate, so there was no complete solubility of them.

The results of microbiological determinations showed a reduction in the total number of forming colony units in all samples, unlike the control sample (Table 5). Only the films with the addition of eucalyptus and chamomile EO did not show any contamination. It is very important that, before use, essential oils to be tested against microbiological contamination and to be purchased only from reliable producers. Even if, in general, they have an antimicrobial effect, they can become contaminated after being obtained.

The obtained results reinforce the possibility that these types of films can be successfully used as packaging materials in the food industry (Figure 3). Depending on the composition, the characteristics of the material may differ, so that it can be used for a wide range of products, with different characteristics.

## 3. Conclusions

Agar-alginate films developed with essential oils incorporated have shown adequate structural properties for use in food packaging applications, especially at 10% EO concentration. All films have totally inhibited or significantly reduced microbial growth, compared to control films, without EO included. Films with 10 and 20% eucalyptus oil showed best performance when considering both structural (thickness, water activity index etc.) and antimicrobial properties. Other well-performing composites were 10% mint and 10% and 20% orange EO films, while the rest of the films had good structural and optical properties and partial reductions of microbiological counts.

The obtained results bring to the fore the advantages of using essential oils in the development of biopolymer films. Even if, for similar polymeric composites, reductions of physico-chemical performances of the films were reported after the addition of such compounds, the results obtained in this case are much better than those of the control sample. Films based on biopolymers have shown their benefits in use. These can be extended and greatly improved by adding essential oils. Easy to obtain, use, and handle, these materials can successfully replace the conventional ones, based on petroleum, difficult to sort and almost impossible to recycle, so harmful for the environment. Future research involves other practical applications of these materials, with time testing of the properties of the films, but also of the packaged products, offering promising avenues for packaging food products.

## 4. Materials and Methods

Agar, alginate, glycerol, and Tween 80 were purchased from Sigma Aldrich (Romanian branch, Bucharest). All the essential oils used—lemon, grapefruit, orange, cinnamon, clove, mint, ginger, eucalypt, and chamomile—were purchased from Carl Roth (Germany). For the development of biopolymer films, the previously developed and tested composition, with some changes, was used [12]. Thus, agar: alginate: glycerol in a ratio of 2: 1: 1 was used for the film-forming solution. After obtaining the film-forming solution followed by stirring for 20 min (90 ± 2 °C and 450 rpm), the solution was cooled to a temperature of 40 °C and 10, respectively 20% *v*/*v* essential oil was added. Then the solutions were poured on silicone support and maintained for 38–42 h, until complete drying, at room temperature (24 ± 3 °C) and rH = 47 ± 3% (Figure 4). A total of 18 samples with the addition of essential oil and one control were obtained. 

After development, the samples were observed to identify the regularity of the edges, the presence of pores and fissures, the uniformity of the film or the degree of homogenization of the substances used. The photos of the developed films are available in Appendix A.

In order to evaluate the physical and optical properties, the thickness, the retraction ratio, the transmittance, the opacity, and the color were tested. The thickness was measured with a Yato micrometer (Shanghai), with an accuracy of 0.001 mm and a test range of 0–25 mm. For the evaluation of the film thickness, five readings were made, in different areas of the surface, and the value noted in Table 3 represents the average. For the retraction ratio (RR) values, the thickness of the film-forming solution (*T*1, 1050 µm), poured on the silicone support immediately after obtaining and the final thickness of the film (*T*2, µm), were taken into account. Thus, the result was calculated by the formula:(1)$$RR,\left(\%\right)=\frac{T1-T2}{T1}*100$$

Transmittance (*T*, %) and absorbance were read spectrophotometrically, using Epoch equipment (BioTek Instruments, Winooski, VT, USA). For the determination, 1 cm × 3 cm film samples were used. The determinations were performed in triplicate. The transmittance was read at a wavelength of 660 nm and the absorbance at 600 nm. The read absorbance was used to determine the opacity, after the following formula was applied:(2)$$Opacity,\;\left(A/mm\right)=\frac{A}{T}$$
where *A*—absorbance and *T*—thickness, mm.

The color was evaluated using the Konica Minolta CR 400 colorimeter. The value taken into account represents the average of at least five readings, made in different areas of the film surface. The parameters considered were L*, a*, b*, and the blank values were L* = 94.12, a* = −5.52, and b* = 9.27.

The method used in previous research [53] was used to determine the humidity. Thus, 3 cm × 3 cm film samples were weighed before and after maintaining for 24 h at a temperature of 110 ± 2 °C. The determination was made in triplicate and the result was calculated with Formula (3):(3)$$MC,\;\left(\%\right)=\frac{W0-W1}{W0}*100$$
where *W*0 represent the sample mass before drying (g) and *W*1 the dried mass (g).

In order to evaluate the solubility, the swelling ratio index was taken into account. This evaluation is important when the material is an edible one and it is intended to be consumed with the product it contains. For determination, 3 × 3 cm film samples were weighed (*M*0), immersed in water at room temperature and maintained for 0.5, 1, 3, 5, 7, 10, 15, 20 min. After this time, the excess water was removed using filter paper, and the samples were re-weighed (*M*1). The determinations were made as single experiment.

The water activity index (*a_w_*) was determined with AquaLab 4TE equipment (Meter Group, München, Germany) and the results were taken into account after at least 5 readings performed in different areas of the film surface. The determinations were made at 25 ± 0.7 °C. 

STAS ASTM D882 ((Standard Test Method for Tensile Properties of Thin Plastic Sheeting) [54] was used to determine the mechanical properties. Thus, for the evaluation of tear resistance and elongation, 1 cm × 10 cm film samples were tested using the ESM Mark 10 universal texturometer loaded with a 5 kN cell. Special grips for thin films and foils were attached. The travel speed was set at 10 mm/min and temperature was 25.5 ± 0.2 °C.

Tensile strength (TS) was calculated according to the Formula (4):(4)$$TS,\;\left(MPa\right)=\frac{F}{S}$$
where *F* is the maximum load (kN) and *S* represent the surface (mm^2^).

The elongation at break represents the ratio between final length and initial length after breakage of the test specimen. It was calculated according to the following formula:(5)$$E,\;\left(\%\right)=\frac{\Delta l}{l}*100$$
where Δ*l* represent the distance between final length and initial length, *l*, in mm.

The mechanical tests were performed in triplicate and the results are noted in Table 3.

Microbiological analysis of the films involved the use of Compact Dry (NISSUI Pharma) type plates, with dehydrated culture medium. Thus, total count (TC), *Escherichia coli* (EC), enterococcus (ETC), coliforms (CF), *Listeria monocytogenes* (LM), *Staphylococcus aureus* (XSA), and yeasts and molds (YM) were tested.

For determination, 1 g of film was solubilized in 9 mL of saline. From the solution thus obtained, 1 mL was poured on the culture medium. After rehydration, the plates were thermostated at 37 degrees for 36 h for TC, EC, ETC, CF, and LM, respectively 72 h for yeasts and molds. 

The microstructure of images and 3D topographies were analyzed using Mountains 9 software (Digital Surf, Lavoisier, France). The roughness was calculated using the operators of the same software, taking into account the highest point, the lowest point and other areas on the surface of the films.

Significant differences of data were evaluated by carrying out one-way analysis of variance (ANOVA) and Tukey’s test at *p* < 0.05. Data analysis was performed using MiniTAB statistics software (MiniTAB Ltd., Coventry, UK).

## 5. Patents

The composition used as a matrix for the incorporation of essential oils was proposed for patenting at the State Office for Inventions and Trademarks in Romania, registered with number A000737/06.12.2021.

## Figures and Tables

**Figure 1 gels-08-00505-f001:**
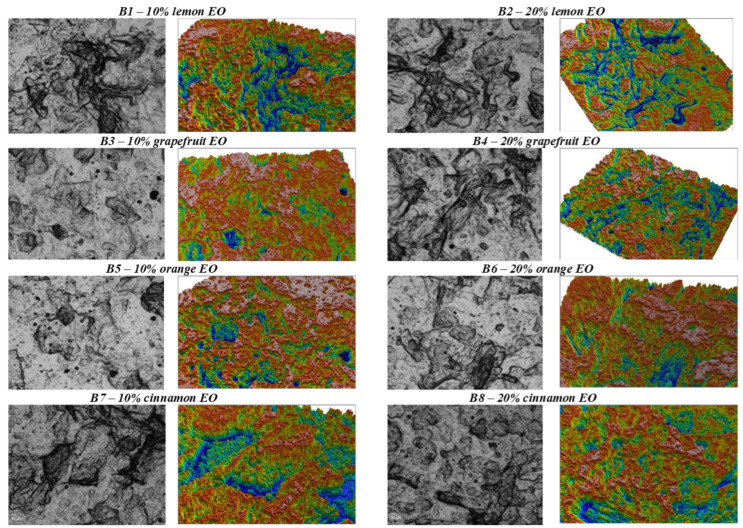
Microstructural and 3D microtopographical aspects of films developed with essential oils included.

**Figure 2 gels-08-00505-f002:**
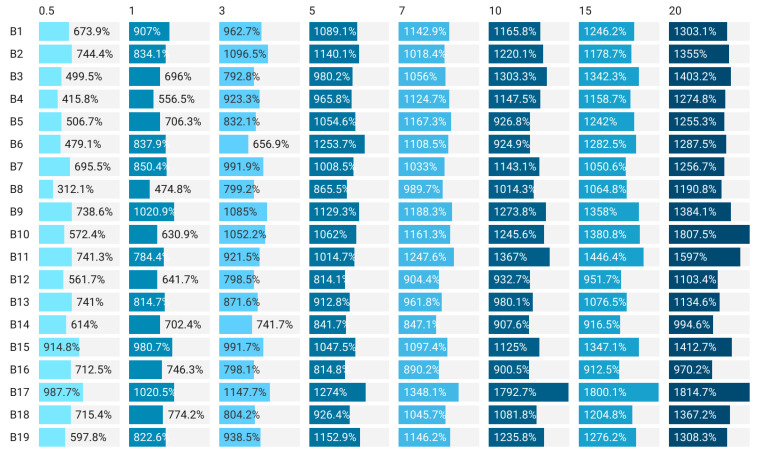
Swelling ratio index of films developed with EO included. B1–B19 sample designation, 0.5–20 represents the time, in minutes, of maintaining the sample in the liquid.

**Figure 3 gels-08-00505-f003:**
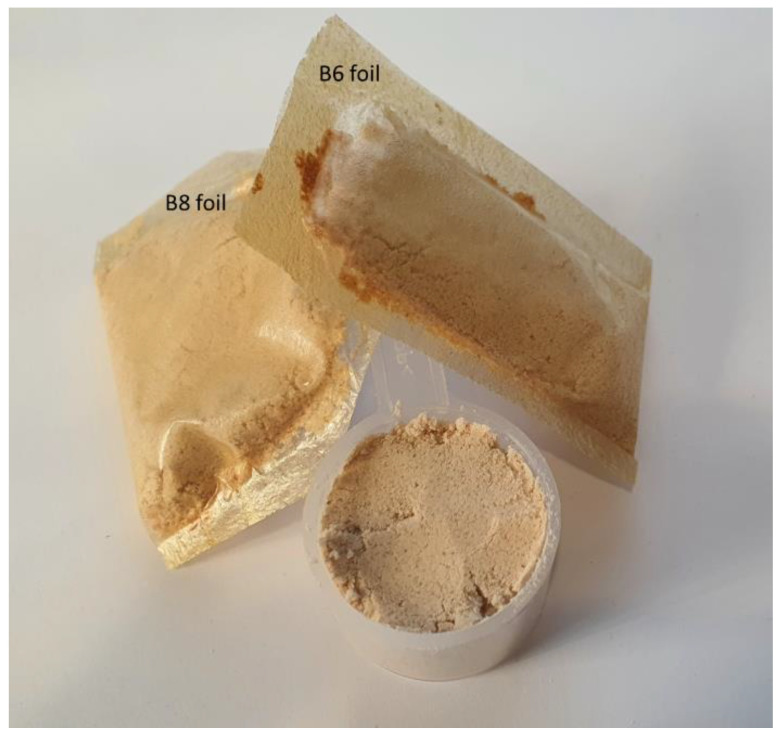
Examples of potential applications of developed films: protein powder packed in biopolymeric foils with orange (**B6**) and cinnamon (**B8**) EO added. Each package contains a measure of protein powder, according to the manufacturer. The packages are glued by hot thermowelding (180 °C, 20 s). The addition of orange and cinnamon EO may improve the sensorial properties of such powders and may sustain beneficial effect on health.

**Figure 4 gels-08-00505-f004:**
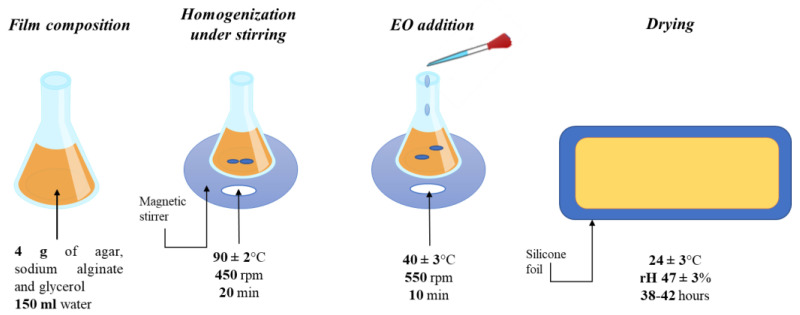
Diagram of schematic preparation process.

**Table 1 gels-08-00505-t001:** Examples of bio-based films incorporating essential oils (EOs) with antimicrobial properties.

Biopolymer	EO	Food Product	Beneficial Effect	References
Starch	Cinnamon	Active packaging	Thermal stability, but porous microstructure	[26]
Chitosan	Lemon	Citrus	Extended shelf -life, improved food storage quality. antimicrobial effect	[27]
Collagen/chitosan	Lemon	Pork meat	Prolonged the shelf life for 21 days, inhibited lipid oxidation, and prevent microbial proliferation	[28]
Pectin	Clove	Fish (bream)	Inhibited the growth of Gram-negative bacteria; the level of lactic acid bacteria remained constant	[29]
Whey protein isolate	Clove	Cheese	Positive effects of the physical-chemical properties of cheese. *E. coli*, *S. aureus*, and *L. monocytogenes* decreased during 60 days of testing	[30]
Alginate/k-carrageenan	Clove	Food packaging	Antimicrobial and antioxidant effect; the addition of EO reduced the mechanical properties of film, but improved the flexibility	[31]
Alginate/clay	Clove, coriander, cinnamon, cumin, caraway, marjoram	Food packaging	Inhibited the growth of Gram-negative bacteria	[32]
Alginate/CMC	Clove	Fish	Without loss of color, odor, texture during storage (16 days)	[33]
Chitosan/pectin/starch	Mint	Food packaging	Antioxidant and antimicrobial effect; improved barrier properties, tensile strength, and thermal stability	[34]
Sodium alginate	Carvacrol	White mushrooms	Improved mechanical properties, water resistance, light barrier property, and antiviral properties	[35]
Chitosan	Eucalyptus	Sliced sausages	Good antimicrobial activity	[36]
Gelatin	Chamomile and peppermint	Edible packaging	The antioxidant activity and bioactivity were improved	[37]
Cellulose	Ginger	Barbecue chicken	Improved spoilage control; the addition of essential oils prolonged the meat shelf life by more than 6 days.	[38]
Chitosan	Ginger	Fresh poultry meat	Reduced coliforms proliferation; the addition of EO showed minimum changes than the product uncoated	[39]
Chitosan	Ginger	Active packaging	Strong antimicrobial activity	[40]
Chitosan/protein	Ginger	Fish	Stored at 4 °C, the shelf life has been extended	[41]

**Table 2 gels-08-00505-t002:** Essential oils displaying antimicrobial properties against relevant microorganism categories and species.

EO	M	TC	*LM*	CF	*SA*	*EC*	References
Cinnamon	√	√	√	√	√	√	[43,44,45,46]
Lemon	√	√		√	√	√	[27,46,47,48]
Grapefruit	√	√		√	√		[48,49,50]
Orange	√	√		√	√	√	[48,50]
Clove		√	√	√	√	√	[29,30,31,32]
Peppermint	√	√	√			√	[34,51]
Eucalyptus		√			√	√	[36]
Chamomile		√			√	√	[37]
Ginger		√	√	√	√	√	[39,52]

M—molds, TC—total count, *LM*—*Listeria monocytogenes*, CF—coliforms, *SA*—*Staphylococcus aureus*, *EC*—*Escherichia coli*.

**Table 3 gels-08-00505-t003:** Structural characteristics of the developed films with essential oils incorporated.

Sample	Thickness, µm	Retraction Ratio, %	Roughness, nm	Tensile Strength, MPa	Elongation, %	Moisture Content, %	Water Activity Index
**B1**	99.80 ^b,c^ ± 2.40	4.95 ^i,j^ ± 2.28	253.10 ^b,c,d,e,f^ ± 2.99	0.101 ^l^ ± 0.10	1.51 ^r^ ± 0.01	11.95 ^e^ ± 0.03	0.33 ^c^ ± 0.01
**B2**	100.60 ^b^ ± 1.85	4.19 ^j^ ± 1.76	333.20 ^a,b^ ± 3.55	0.098 ^l^ ± 0.05	1.91 ^q^ ± 0.05	11.24 ^f^ ± 0.01	0.29 ^d^ ± 0.04
**B3**	88.80 ^f,g^ ± 1.47	15.43 ^e,f^ ± 1.40	366.20 ^a^ ± 3.03	0.231 ^e^ ± 0.05	14.97 ^d^ ± 0.20	13.24 ^c^ ± 0.22	0.30 ^c,d^ ± 0.01
**B4**	90.80 ^e,f^ ± 1.17	13.52 ^f,g^ ± 1.11	243.70 ^b,c,d,e,f,g^ ± 4.16	0.172 ^f^ ± 0.01	11.04 ^g^ ± 0.33	11.27 ^f^ ± 0.61	0.30 ^c,d^ ± 0.01
**B5**	81.00 ^h,i^ ± 2.68	22.86 ^c,d^ ± 2.55	255.00 ^b,c,d,e,f^ ± 1.03	0.133 ^l^ ± 0.01	7.10 ^j^ ± 0.15	13.73 ^b^ ±0.13	0.30 ^c,d^ ± 0.08
**B6**	93.60 ^d,e^ ± 1.85	10.86 ^g,h^ ± 1.76	304.87 ^a,b,c,d^ ± 0.85	0.146 ^g^ ± 0.01	5.37 ^n^ ± 0.11	13.53 ^b^ ±0.29	0.30 ^c,d^ ± 0.03
**B7**	98.80 ^b,c^ ± 1.94	5.90 ^i,j^ ± 1.84	307.47 ^a,b,c,d^ ± 1.96	0.137 ^i^ ± 0.01	6.92 ^k^ ± 0.10	13.13 ^c^ ± 0.14	0.33 ^c^ ± 0.01
**B8**	101.00 ^b^ ± 0.89	3.81 ^j^ ± 0.85	308.97 ^a,b,c,d^ ± 0.47	0.130 ^k^ ± 0.05	8.33 ^i^ ± 0.21	12.74 ^d^ ±0.68	0.30 ^c,d^ ± 0.02
**B9**	91.60 ^e,f^ ± 1.50	12.76 ^f,g^ ± 1.42	293.67 ^a,b,c,d^ ± 0.59	0.093 ^m^ ± 0.01	3.74 ^p^ ±0.07	13.24 ^c^ ± 0.76	0.29 ^c,d^ ± 0.01
**B10**	98.80 ^a,b^ ± 1.67	4.76 ^i,j^ ± 1.59	314.33 ^a,b,c^ ± 0.51	0.142 ^h^ ± 0.01	13.87 ^e^ ± 0.07	13.04 ^c^ ± 0.75	0.27 ^d^ ± 0.01
**B11**	59.40 ^j^ ± 1.36	43.43 ^b^ ± 1.29	216.17 ^d,e,f,g^ ± 0.68	0.274 ^c^ ± 0.05	17.45 ^b^ ± 0.18	11.26 ^f^ ± 0.25	0.52 ^b^ ± 0.02
**B12**	76.80 ^i^ ± 1.60	26.86 ^c^ ± 1.52	169.23 ^f,g^ ± 0.57	0.288 ^a^ ± 0.15	16.42 ^c^ ± 0.02	10.34 ^g^ ±0.20	0.52 ^a,b^ ± 0.02
**B13**	101.20 ^b^ ± 1.33	3.62 ^j^ ± 1.26	248.37 ^b,c,d,e,f^ ± 4.46	0.261 ^d^ ± 0.10	26.11 ^a^ ± 0.03	11.73 ^e^ ± 0.19	0.56 ^a^ ± 0.03
**B14**	102.00 ^a,b^ ±1.10	2.86 ^j^ ± 1.04	288.27 ^a,b,c,d,e^ ± 4.02	*	*	8.73 ^h^ ± 0.20	0.56 ^a^ ± 0.02
**B15**	92.00 ^d,e,f^ ± 0.89	12.38 ^f,g,h^ ± 0.85	230.60 ^c,d,e,f,g^ ± 3.04	0.175 ^g^ ± 0.10	8.93 ^h^ ± 0.02	8.51 ^h^ ± 0.43	0.56 ^a^ ± 0.02
**B16**	98.80 ^b,c^ ± 1.33	5.90 ^i,j^ ± 1.26	192.03 ^e,f,g^ ± 3.61	0.148 ^g^ ± 0.05	6.71 ^l^ ± 0.06	19.34 ^a^ ±0.11	0.55 ^a,b^ ± 0.01
**B17**	53.00 ^k^ ± 1.67	49.52 ^a^ ± 1.59	146.90 ^g^ ± 0.57	0.282 ^b^ ± 0.02	12.98 ^f^ ± 0.01	2.43 ^j^ ± 0.19	0.53 ^a,b^ ± 0.02
**B18**	96.20 ^c,d^ ± 1.17	8.38 ^h,i^ ± 1.11	222.60 ^c,d,e,f,g^ ± 3.63	0.131 ^k^ ± 0.10	6.44 ^m^ ± 0.01	4.54 ^i^ ± 0.36	0.53 ^a,b^ ± 0.01
**B19**	85.20 ^g,h^ ± 1.94	18.86 ^d,e^ ± 1.84	176.93 ^f,g^ ± 1.17	0.135 ^i.j^ ± 0.10	5.04 ^o^ ± 0.02	11.44 ^f^ ± 0.18	0.28 ^d^ ± 0.02

* The testing could not be carried out because the sample could not be sized according to STAS ASTM D882. Means that do not share a letter are significantly different. a–j, Significance level α = 0.05.

**Table 4 gels-08-00505-t004:** Optical properties of films with essential oils incorporated.

Sample	Transmittance, %	Opacity, A × mm^−1^	Color		
			L*	a*	b*
**B1**	49.40 ^i^ ± 0.10	3.31 ^e^ ± 0.04	88.79 ^b,c^ ± 0.18	−0.49 ^a^ ± 0.02	10.76 ^e^ ± 0.31
**B2**	40.66 ^k^ ± 0.75	3.59 ^b^ ± 0.12	88.99 ^b,c^ ± 0.16	−0.42 ^a^ ± 0.24	10.61 ^e^ ± 0.29
**B3**	51.00 ^g^ ± 0.20	3.21 ^f^ ± 0.14	88.30 ^b,c,d^ ± 0.39	−0.54 ^a^ ± 0.02	11.33 ^d,e^ ± 0.48
**B4**	52.60 ^e^ ± 0.20	3.49 ^c^ ± 0.22	88.87 ^b,c^ ± 0.96	−0.61 ^a^ ± 0.05	10.72 ^e^ ± 0.53
**B5**	53.10 ^e^ ± 0.10	3.47 ^c^ ± 0.15	88.88 ^b,c^ ± 0.73	−0.51 ^a^ ± 0.86	9.35 ^e^ ± 0.65
**B6**	51.73 ^f^ ± 0.15	3.50 ^c^ ± 0.10	89.17 ^b^ ± 0.63	−0.48 ^a^ ± 0.15	9.85 ^e^ ± 0.73
**B7**	50.16 ^h^ ± 0.11	3.26 ^e,f^ ± 0.20	89.17 ^b^ ± 0.30	−0.48 ^a^ ± 0.16	10.03 ^e^ ± 0.28
**B8**	50.83 ^g,h^ ± 0.11	3.02 ^h^ ± 0.09	88.58 ^b,c^ ± 0.92	−0.49 ^a^ ± 0.02	11.21 ^d,e^ ± 0.83
**B9**	50.43 ^g,h^ ± 0.15	3.39 ^d^ ± 0.08	88.50 ^b,c,d^ ± 0.93	−0.47 ^a^ ± 0.03	11.16 ^d,e^ ± 1.02
**B10**	52.73 ^e^ ± 0.11	3.03 ^g,h^ ± 0.16	88.31 ^b,c,d^ ± 0.68	−0.52 ^a^ ± 0.01	10.73 ^e^ ± 0.65
**B11**	71.63 ^c^ ± 0.11	2.53 ^j^ ± 0.08	92.25 ^a^ ± 0.79	−5.78 ^c^ ± 0.07	13.70 ^b,c^ ± 0.53
**B12**	73.56 ^b^ ± 0.12	1.88 ^l^ ± 0.21	92.18 ^a^ ± 0.15	−5.79 ^c^ ± 0.02	12.84 ^c,d^ ± 0.52
**B13**	70.01 ^a^ ± 0.03	3.99 ^a^ ± 0.12	91.00 ^a^ ± 0.64	−5.76 ^c^ ± 0.01	15.26 ^b^ ± 0.94
**B14**	18.33 ^l^ ± 0.21	3.08 ^g^ ± 0.12	92.07 ^a^ ± 0.28	−5.85 ^c^ ± 0.03	13.61 ^b,c^ ± 0.37
**B15**	41.30 ^k^ ± 0.10	1.52 ^m^ ± 0.17	92.19 ^a^ ± 0.28	−5.83 ^c^ ± 0.03	13.11 ^c,d^ ± 0.42
**B16**	46.60 ^j^ ± 0.20	1.15 ^n^ ± 0.12	91.83 ^a^ ± 0.15	−5.76 ^c^ ± 0.08	14.31 ^b,c^ ± 1.17
**B17**	73.5 ^b^ ± 0.20	2.87 ^i^ ± 0.23	92.09 ^a^ ± 0.44	−5.91 ^c^ ± 0.10	12.84 ^c,d^ ± 0.67
**B18**	40.73 ^k^ ± 0.11	1.5 ^m^ ± 0.13	86.97 ^d^ ± 1.09	−5.18 ^b^ ± 0.22	21.97 ^a^ ± 1.09
**B19**	68.83 ^d^ ± 0.11	1.97 ^k^ ± 0.11	87.63 ^c,d^ ± 0.41	−0.62 ^a^ ± 0.02	10.32 ^e^ ± 0.56

***** Means that do not share a letter are significantly different. a–n, Significance level α = 0.05.

**Table 5 gels-08-00505-t005:** Microbiological assessment (ufc/g) of films developed with essential oils incorporated for relevant microorganism categories and species.

Sample	TC	EC	ETC	CF	YM	X-SA	LM
**B1**	1	-	-	1	-	-	-
**B2**	-	-	-	-	-	-	-
**B3**	15	-	-	-	-	-	-
**B4**	21	-	-	-	-	1	-
**B5**	-	-	-	-	-	-	-
**B6**	7	-	-	-	-	-	-
**B7**	2	-	-	-	-	-	-
**B8**	-	-	-	-	-	-	-
**B9**	13	-	-	2	-	-	-
**B10**	1	-	-	-	-	-	-
**B11**	-	-	-	2	-	-	-
**B12**	8	-	-	-	-	-	-
**B13**	19	-	-	-	-	1	-
**B14**	6	-	-	-	-	-	-
**B15**	-	-	-	-	-	-	-
**B16**	-	-	-	-	-	-	-
**B17**	-	-	-	-	-	-	-
**B18**	-	-	-	-	-	-	-
**B19**	28	-	-	-	-	-	-

TC—total count, EC—*Escherichia coli*, ETC—*Enterococcus*, CF—*coliforms*, YM—yeasts and molds, X-SA—*Staphylococcus aureus*, LM—*Listeria monocytogenes*.

## Data Availability

Not applicable.

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
