# Peer review of "The Future Packaging of the Food Industry: The Development and Characterization of Innovative Biobased Materials with Essential Oils Added"

_gels, 2022, doi:10.3390/gels8080505_

Round 1

Reviewer 1 Report

The research work related to the development of edible and biodegradable packaging material fortified with essential oils is very interesting and has industrial importance. Authors have conducted analyses in a novel way. However, it needs a major revision. I have provided comments in the attched PDF file. Moreover english corrections and intended meaning of sentences must be carefully resvised. In my openion manuscript shold be accepted after major revision. 

Author Response

The research work related to the development of edible and biodegradable packaging material fortified with essential oils is very interesting and has industrial importance. Authors have conducted analyses in a novel way. However, it needs a major revision. I have provided comments in the attched PDF file. Moreover english corrections and intended meaning of sentences must be carefully resvised. In my openion manuscript shold be accepted after major revision.

                Response: Thank you for the review. The manuscript was improved, according to the indications of the three reviewers.

                Title should be revised to display the researched work. This title is suitable for a review paper.

               Response: The title was changed, thank you.

               Abstract must be rewritten to present the results of the study. It looks like a general information. Please make it more technical and catchy for readers.

               Response: the abstract was improved. We added information and we hope that it is ok now (lines 18-22).

               Please revise the introduction. It needs major changes to present sequential literature for better understanding of the research you have conducted.

               Response: We correct the errors and we added new references.

               Title of the table 3 must be revised to reflect all the analyses.

               Response: we modified, thank you.

               The figure caption is incomplete. More elaborated text is required. Authors need to indicate which figure is Microstructures and 3D microtopographies???

               Response: we added new information, thank you. First image is microstructure and second one represents a 3D topography.

               Please provide the full information of sample abbreviation in the figure caption.

               Response: We have modified the figure in order to be easier to follow.

               Table 4 caption and footnote are not correctly presented. Please complete.

               Response: We completed, thank you.

               Authors should seriously revise the table caption and footnote to present all the abbreviations.

               Response: we completed it, thank you.

               Grammatical errors must be corrected.

               Response: we modified the figure caption (lines 261-267).

               Conclusion must be the reflection of technical. The optimum levels of EO and packaging material is missing. Please revise and make it conclusive rather than it looks like a passage.

               Response: We complete the conclusion part (lines 269-280).

               “.....followed by stirring....”; Better to use degree as symbol.

               Response: we modified, thank you.

Reviewer 2 Report

This study tested the modification of the properties of the previously developed biopolymeric material, concerning about physical, optical, microbiological or solubility properties. All the films with the addition of essential oil in the composition showed improved properties compared to the control sample. This study is the focus of many researchers at present. This study is relatively simple, involves few mechanisms, or the related mechanism is thin. And there are still some problems in the current manuscript that need to be clarified.

1. Abstract, results and highlights are all qualitative expressions and lack of quantitative data.

2. What is the release mechanism of essential oils in those films?

3. The table should be a three-line table.

4. Correct the header for Table. 3.

5. Please add a ruler for the Fig. 1.

6. Please check the manuscript carefully, refine the language and correct the mistakes.

Author Response

This study tested the modification of the properties of the previously developed biopolymeric material, concerning about physical, optical, microbiological or solubility properties. All the films with the addition of essential oil in the composition showed improved properties compared to the control sample. This study is the focus of many researchers at present. This study is relatively simple, involves few mechanisms, or the related mechanism is thin. And there are still some problems in the current manuscript that need to be clarified.

Abstract, results and highlights are all qualitative expressions and lack of quantitative data.

Response: Thank you for your review. The abstract was added with quantitative information (lines 18-22), we added mechanical tests (Table 3 and lines 139-159), and conclusions part has been improved (lines 269-281).

               What is the release mechanism of essential oils in those films?

Response: The actual mechanisms of oil release is an ongoing investigation of the authors and will be presented in another original research paper.

               The table should be a three-line table.

Response: The tables were updated, thank you.

               Correct the header for Table. 3.

Response: We kindly ask the reviewer to explicit the corrections that should be performed. Thank you.

               Please add a ruler for the Fig. 1.

Response: A scale was added.

               Please check the manuscript carefully, refine the language and correct the mistakes.

Response: We corrected the errors and we refined the English language, thank you.

Reviewer 3 Report

In this manuscript (gels-1855949-pee) entitled “Is active packaging the future of the food industry? The advantages of using essential oils in the development of biobased-materials”, authors have reported the preparation and research of biopolymer films modified with different essential oils for packaging materials. The manuscript is well-organized and the conclusion is supported by the experiment and results. Therefore, this reviewer would suggest an acceptance after addressing the following moderate issues.

1.      Please add a scale in Fig. 1 and provide the photos of each sample. Please use the schematic diagram to show the sample preparation process.

2.      For the statement that “With the exception of the film with the addition of ginger essential oil, which broke during drying and showed extremely low flexibility, all other films were very flexible and allowed multiple bending” in page 4, line 117-119, please give the specific experimental results that characterize the flexibility of the samples.

3.      For scientific expression, three line tables should be applied.

4.      For the statement that “The sample with 10% essential oil of mint showed high adhesiveness and can be used as a self-adhesive film” in page 4, line 120-121, there is no relevant experimental data in the paper.

5.      Some lines in Fig. 2 are illegible, such as B10, B17 and B19.

6.      The maximum difference in sample thickness should be 43.20 μm instead of 46.20 μm (page 9, line 144). The thickness difference between B13 and B14 should be 0.8 μm instead of 8.8 μm (page 9, line 148). The thickness difference between B1 and B2 is also 0.8 μm, which is also the smallest thickness difference between the samples.

7.      The expression that “Even so, with the exception of the films with chamomile oil in the composition (B17, B18), the values obtained for the other samples vary slightly (± 2.29 %) in contrast to the control sample (B19)” in page 9, line 159-161 is not rigorous, and there is also a big gap between the moisture content of B16 and B19.

8.      The water activity index of clove EO is similar to that of the control, rather than mint EO (page 10, line 167-169).

9.      Is the transmittance of some samples correct, such as B1 (0.33 ± 0.94)%? Is the transmittance of all samples really so small, less than 0.5%?

10.  In addition to physical, optical, microbial or solubility properties, the mechanical properties of biopolymer films used in packaging materials are also important considerations. Please supplement the mechanical property test of each sample, such as tensile test.

11.  When generally introducing the background and giving examples of packaging materials, some recent articles should be considers: Packaging and degradability properties of polyvinyl alcohol/gelatin nanocomposite films filled water hyacinth cellulose nanocrystals; Development and characterization of food packaging bioplastic film from cocoa pod husk cellulose incorporated with sugarcane bagasse fibre; Electrospun Functional Materials toward Food Packaging Applications: A Review.

12.  W0 and W1 are used in the test of the moisture content and swelling ratio index of the sample, but their meanings of W0 and W1 are different (page 12), which is not appropriate. In addition, the numbers in all formulas should be subscripted. It is suggested to add “(EO)” after “essential oils” in the caption of Table 1.

13.  There are some errors in the paper, such as “Almost 80% from globally produced agar it is used for human consumption” in page 2, line 57; “in 2009” in page 2, line 63; “4°C” in page 3, line 85; “Thus, total count (TC), Escherichia coli (EC), enterococcus (ETC), coliforms (CF), Listeria monocytogenes (LM), Staphylococcus aureus (XSA), and yeasts and molds (YM)” in page 12, line 278-280. Please check the full paper carefully.

Author Response

In this manuscript (gels-1855949-pee) entitled “Is active packaging the future of the food industry? The advantages of using essential oils in the development of biobased-materials”, authors have reported the preparation and research of biopolymer films modified with different essential oils for packaging materials. The manuscript is well-organized and the conclusion is supported by the experiment and results. Therefore, this reviewer would suggest an acceptance after addressing the following moderate issues.

Response: Thank you. We appreciate your effort and time.

Please add a scale in Fig. 1 and provide the photos of each sample. Please use the schematic diagram to show the sample preparation process.

Response: A scale was added and photos were added as supplementary material, while the preparation process was represented in Fig 4.

For the statement that “With the exception of the film with the addition of ginger essential oil, which broke during drying and showed extremely low flexibility, all other films were very flexible and allowed multiple bending” in page 4, line 117-119, please give the specific experimental results that characterize the flexibility of the samples.

For scientific expression, three-line tables should be applied.

Response: The table were modified, thank you!

For the statement that “The sample with 10% essential oil of mint showed high adhesiveness and can be used as a self-adhesive film” in page 4, line 120-121, there is no relevant experimental data in the paper.

Some lines in Fig. 2 are illegible, such as B10, B17 and B19.

Response: The figure was modified.

The maximum difference in sample thickness should be 43.20 μm instead of 46.20 μm (page 9, line 144). The thickness difference between B13 and B14 should be 0.8 μm instead of 8.8 μm (page 9, line 148). The thickness difference between B1 and B2 is also 0.8 μm, which is also the smallest thickness difference between the samples.

Response: All data were corrected; we apologize for the errors.

The expression that “Even so, with the exception of the films with chamomile oil in the composition (B17, B18), the values obtained for the other samples vary slightly (± 2.29 %) in contrast to the control sample (B19)” in page 9, line 159-161 is not rigorous, and there is also a big gap between the moisture content of B16 and B19.

Response: The text was reviewed.

The water activity index of clove EO is similar to that of the control, rather than mint EO (page 10, line 167-169).

Response: The text was reviewed, thank you.

Is the transmittance of some samples correct, such as B1 (0.33 ± 0.94) %? Is the transmittance of all samples really so small, less than 0.5%?

Response: we corrected the results, please excuse our errors. Thank you for your notice.

In addition to physical, optical, microbial or solubility properties, the mechanical properties of biopolymer films used in packaging materials are also important considerations. Please supplement the mechanical property test of each sample, such as tensile test.

Response: we performed the mechanical tests, such as tensile strength and elongation. The results are noted in Table 3 and the discussions can be seen at lines 139-160.

When generally introducing the background and giving examples of packaging materials, some recent articles should be considers: Packaging and degradability properties of polyvinyl alcohol/gelatin nanocomposite films filled water hyacinth cellulose nanocrystals; Development and characterization of food packaging bioplastic film from cocoa pod husk cellulose incorporated with sugarcane bagasse fibre; Electrospun Functional Materials toward Food Packaging Applications: A Review.

Response: The supporting material were added (references 2 and 22), we gratefully thank the reviewer.

W0 and W1 are used in the test of the moisture content and swelling ratio index of the sample, but their meanings of W0 and W1 are different (page 12), which is not appropriate. In addition, the numbers in all formulas should be subscripted. It is suggested to add “(EO)” after “essential oils” in the caption of Table 1.

Response: The abbreviations were reviewed, thank you.

There are some errors in the paper, such as “Almost 80% from globally produced agar it is used for human consumption” in page 2, line 57; “in 2009” in page 2, line 63; “4°C” in page 3, line 85; “Thus, total count (TC), Escherichia coli (EC), enterococcus (ETC), coliforms (CF), Listeria monocytogenes (LM), Staphylococcus aureus (XSA), and yeasts and molds (YM)” in page 12, line 278-280. Please check the full paper carefully.

Response: The text was accordingly revised, thank you.

Round 2

Reviewer 3 Report

Authors have addressed all the issues. An acceptance is suggested.